# Measurement-induced collective vibrational quantum coherence under spontaneous Raman scattering in a liquid

Valeria Vento [1,3], Santiago Tarrago Velez[1,2,3], Anna Pogrebna[1] & Christophe Galland [1] ✉

Spontaneous vibrational Raman scattering is a ubiquitous form of light–matter interaction whose description necessitates quantization of the electromagnetic field. It is usually considered as an incoherent process because the scattered field lacks any predictable phase relationship with the incoming field. When probing an ensemble of molecules, the question therefore arises: What quantum state should be used to describe the molecular ensemble following spontaneous Stokes scattering? We experimentally address this question by measuring time-resolved Stokes–anti-Stokes two-photon coincidences on a molecular liquid consisting of several sub-ensembles with slightly different vibrational frequencies. When spontaneously scattered Stokes photons and subsequent anti-Stokes photons are detected into a single spatiotemporal mode, the observed dynamics is inconsistent with a statistical mixture of individually excited molecules. Instead, we show that the data are reproduced if Stokes–anti-Stokes correlations are mediated by a collective vibrational quantum, i.e. a coherent superposition of all molecules interacting with light. Our results demonstrate that the degree of coherence in the vibrational state of the liquid is not an intrinsic property of the material system, but rather depends on the optical excitation and detection geometry.

Raman scattering was first reported in 1928[1] and with the advent of laser sources it has become an essential tool for probing and understanding the vibrational structure of organic and inorganic matter. In a majority of experiments, a semi-classical model of light–matter interaction is sufficient to interpret the results of Raman spectroscopy. For example, the intensity asymmetry between Stokes and anti-Stokes scattering is obtained by quantizing the vibrational modes of each individual molecule. A full quantum theory of Raman scattering was developed in the 70s and 80s[2–7], and its predictions were tested, e.g., by measuring intensity fluctuations in stimulated Raman scattering[8–10].

Following a pioneering work by I. Walmsley and coworkers in 2011[11], more recent experiments have used time-correlated single photon counting to evidence non-classical intensity correlations between light fields interacting with the same phonon mode via Raman scattering, with potential applications in ultrafast quantum information processing[12–17], novel forms of spectroscopy[18,19], and the generation of non-classical states of light[20]. These experimental results have spurred further theoretical developments to understand how the Raman process leads to photonic correlations mediated by a phononic excitation[21–24], how the experimental geometry impacts the photon statistics of the Stokes field[25], and how the coupling of a Raman-active mode to a nanocavity modifies the dynamics of the system[26–28].

To our knowledge, and despite this recent experimental progress, there has been no direct measurement of the nature of the vibrational quantum state generated in an ensemble of molecules in the liquid phase. Since molecule–molecule interactions in a liquid are in general

[1]Institute of Physics, École polytechnique fédérale de Lausanne (EPFL), CH-1015 Lausanne, Switzerland. [2]Present address: Department of Physics, Technical University of Denmark, Kongens Lyngby, Denmark. [3]These authors contributed equally: Valeria Vento, Santiago Tarrago Velez. ✉e-mail: chris.galland@epfl.ch

incoherent and only contribute to vibrational relaxation[29], they cannot generate spatial coherence over mesoscopic length scales—in contrast to crystalline materials, in which near-field Raman scattering experiments[30–32] have deduced phonon coherence lengths on the order of tens of nanometers. Accordingly, most reference texts assume that coherence among different molecules can only be imposed by external driving, e.g. with the beat note between two strong laser fields as in coherent anti-Stokes Raman scattering (CARS)[33]. Quantum coherence among different molecules following spontaneous Raman scattering in a dense molecular liquid has often been neglected[34,35], implicitly assuming that the resulting collective vibrational state is a statistical mixture of individually excited molecules. We note that when studying an ensemble of identical molecules, neither the temporal coherence of the Stokes field[19,36] nor the presence of Stokes–anti-Stokes coincidences[37,38] provide direct information about the collective coherence possibly existing among the molecules—which is why the authors from Ref. 38 could describe their experiment in terms of single-molecule scattering events.

In this work, we demonstrate that spontaneous vibrational Raman scattering in the liquid phase and at room temperature can also induce a collective vibrational state, i.e. a quantum superposition of a macroscopic number of individually excited molecules. To this end, we use liquid carbon disulfide ($CS_2$) that naturally contains a few distinct molecular sub-ensembles distinguished by their initial vibrational state and isotope content, and having slightly different vibration frequencies (Fig. 1). We measure time-resolved two-photon Stokes–anti-Stokes correlations mediated by the creation and annihilation of a vibrational quantum in the symmetric stretch mode of the molecules, and observe revivals after several picoseconds. These quantum beats are signatures of a coherent superposition where a single vibrational quantum is shared between all molecules involved in Raman scattering and belonging to a spatial mode selected by the measurement geometry. Our results are consistent with the numerous works on emissive quantum memories using atomic ensembles[39] where collective quantum coherence can be induced by post-selection upon single photon

detection—a key concept underlying the DLCZ proposal for quantum repeaters[40]. As proposed in the context of cavity optomechanics[41], our data are also consistent with the emergence of mode entanglement between molecular sub-ensembles upon single photon detection.

Our experiment, performed in a regime of very low Raman scattering probability (about one per hundred pulses), departs from the vast body of literature reporting beat notes under stimulated Raman scattering and other nonlinear spectroscopic data, where classical coherence is imposed by the nonlinear interaction with the excitation beams[42–44]. Our results are also distinct from the quantum beats observed when different energy levels of the same molecule are coherently excited[45–47], since each single molecule within our ensemble vibrates at a single frequency, and only the collective excitation displays quantum beats.

## Results

Figure 1 shows the Raman spectrum of liquid $CS_2$ (anhydrous, ≥99%, Sigma Aldrich) at room temperature, zooming in on the Raman shift of the symmetric stretch mode $\nu_1$, acquired under 780 nm continuous-wave excitation with a high-resolution spectrometer. Note that $CS_2$ has no electronic transition at visible or near-infrared frequencies so that all measurements reported here correspond to far off-resonance excitation. The vibrational transitions associated to the four main peaks in Fig. 1 are assigned following Refs. 48–50, and are labeled according to the initial (at thermal equilibrium) and final (after Stokes scattering) occupation numbers of the stretching ($\nu_1$) and bending ($\nu_2$) vibrational modes.

We recognize the pure $\nu_1$ bands of the two dominant isotopes $CS_2^{32}$ and $CS^{32}S^{34}$ at 659.6 cm$^{-1}$ and 650.1 cm$^{-1}$, respectively, and two hot bands of $CS_2^{32}$ at 657 cm$^{-1}$ and 651.7 cm$^{-1}$. In the following, these four dominant vibrational transitions are simply labeled with $i = 1, 2, 3, 4$ in order of decreasing Raman shift. The pure $\nu_1$ band of $CS^{32}S^{33}$ (barely distinguishable around 655 cm$^{-1}$) and other hot bands of the three isotopes $CS_2^{32}$, $CS^{32}S^{33}$ and $CS^{32}S^{34}$ contribute to the background used in our fit and marked as dashed lines.

After background subtraction we fitted the Raman spectrum in frequency space with the sum of four Voigt functions: for each peak, we assume that the Raman line shape is Lorentzian, while the Gaussian contribution is fixed by the instrument response function. The latter is estimated by measuring the spectrum of the attenuated excitation laser, which is well fit with a Gaussian of ~0.02 nm FWHM (orange shaded peak), in agreement with the nominal system resolution. The free fit parameters are the central frequencies of the four Raman peaks and the FWHMs $\Delta\nu_i$ of three modes ($i = 1, 2, 3$)—the pure $\nu_1$ vibrations of both isotopes are supposed to have the same linewidth, $\Delta\nu_1 = \Delta\nu_4$. The relative intensities among all peaks are fixed by temperature and isotope abundance following Ref. 48. From the Raman shifts, we obtain the four vibrational frequencies: 20.04, 19.96, 19.81 and 19.76 THz. From the Lorentzian width, we infer the effective coherence time $T_2 = (\pi\Delta\nu)^{-1}$ of each vibration in our measurement geometry: 15.71, 18.38, 6.77 and 15.71 ps, respectively.

To investigate the collective vibrational quantum state generated upon Stokes scattering, we use the technique introduced in Ref. 51, as we explain in the Methods; a simplified description of the experimental setup is provided in Fig. 2, while a more detailed scheme can be found in the Supplementary Methods. A first near-infrared laser pulse (~200 fs pulse duration, 80 MHz repetition rate) generates a two-mode photon-vibration squeezed state via spontaneous Stokes scattering. After this *write* pulse, a *read* pulse centered at a different wavelength is used to probe the vibrational mode through anti-Stokes scattering after a variable time delay $t$. The normalized Stokes–anti-Stokes correlation function $g_{S,A}^{(2)}(t)$ reflects the decay of the vibrational excitation heralded by the detection of a Stokes photon[52,53]. In the limit of low scattering probability and absent any background emission nor noise, the strength of vibration-mediated correlation is upper-bounded by

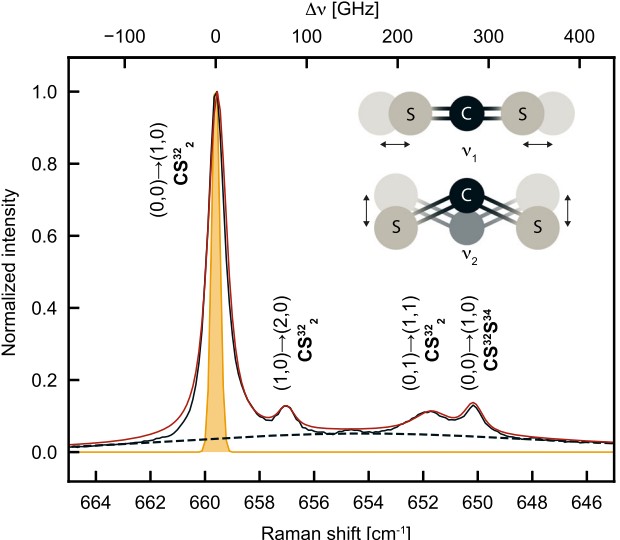

**Fig. 1 | Continuous-wave high-resolution Raman spectrum of liquid CS₂.** The black solid line represents the normalized Stokes intensity measured under 780 nm laser excitation. Each peak is associated with a molecular transition of a particular CS₂ isotope. In the notation $(n_1^{init.}, n_2^{init.}) \rightarrow (n_1^{fin.}, n_2^{fin.})$, $n_1$ and $n_2$ are the quantum numbers of the symmetric stretching mode $\nu_1(\Sigma_g^+)$ and of the two-fold degenerate bending mode $\nu_2(\Pi_u)$, respectively, as sketched in the inset. After subtracting a background (dashed line) the spectrum is fitted in frequency space with the sum of four Voigt functions (red line) with fixed Gaussian contribution (orange profile) as described in the text.

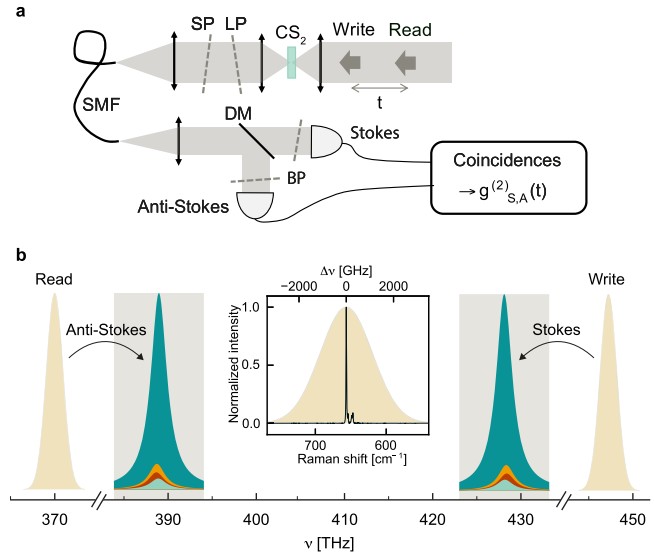

**Fig. 2 | Simplified experimental layout and frequency-domain schematic.**
**a** Simplified sketch of the experimental setup. Liquid $CS_2$ is held between two objective lenses (numerical aperture: 0.8) in a quartz cuvette. Fiber-coupled single photon avalanche photodiodes are connected to a coincidence counter to measure $g_{S,A}^{(2)}$. SP short-pass filter; LP long-pass filter; BP band-pass filter; DM dichroic mirror, SMF single mode fiber. **b** Frequency-domain schematic of experimentally relevant optical fields. Only the Stokes signal from the write pulse and the anti-Stokes signal from the read pulse are transmitted through the filters (gray area) to their respective detectors. Each Raman peak inherits the linewidth of the excitation pulse (~200 fs duration) so that adjacent vibrational modes (Fig. 1) are no longer distinguishable. Inset: comparison between the excitation pulse width (shaded curve) and the continuous-wave Raman spectrum (black line).

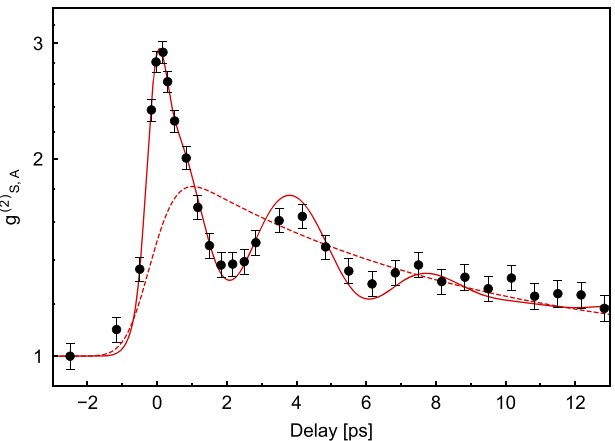

**Fig. 3 | Time-resolved Stokes–anti-Stokes correlations.** Full circles represent measured data points while the solid red line indicates the model prediction (up to a vertical scaling factor). Only Raman photons co-polarized with their respective laser pulses were selected. Free fitting parameters entering the model curve are a scaling amplitude $A_{tot} = 1.67$ for the oscillations and a rising time $T_{rise} = 0.66$ ps. To account for electronic four-wave mixing we add a Gaussian peak centered at zero delay of amplitude $A_g = 1.06$. The dotted red line represents the multi-exponential decay that would result from a statistical mixture of single vibrating molecules.

$g_{S,A}^{(2)} < 1 + 1/n_{th} \simeq 26$, where the thermal occupancy of the vibrational mode is $n_{th} \simeq 0.04$ in our system at room-temperature.

The sample is studied in transmission to fulfill momentum conservation, where the Stokes–anti-Stokes process is mediated by a collective vibration with vanishing momentum. The Raman signal is collected into a single-mode optical fiber whose back-propagated image overlaps with the focused laser beams to define a single spatial mode inside the sample. After spatial filtering through the fiber, the Stokes and anti-Stokes photons from the first and second pulses, respectively, are separated based on their non-overlapping spectra and are individually detected.

Figure 3 shows the measured $g_{S,A}^{(2)}(t)$ plotted as a function of time delay between Stokes and anti-Stokes processes. When write and read pulses temporally overlap, virtual electronic processes contribute to four-wave mixing (FWM) and generate photon pairs at any frequencies satisfying energy conservation (phase matching is highly relaxed in our strongly focused geometry). We used crossed-polarized laser pulses to minimize the relative contribution of electronic FWM to the overall signal. Beyond $t \simeq 1$ ps, instantaneous contributions to FWM have vanished and $g_{S,A}^{(2)}(t) > 1$ is a signature of intensity correlations between Stokes and anti-Stokes fields mediated by molecular vibrations. The red dashed curve is a tentative fit with a multi-exponential decay whose relative amplitudes and time constants are extracted from Fig. 1. This decay would result from describing the vibrational state as a statistical mixture with one molecule excited at random for each coincidence count. While this model matches the overall damping of correlations, it fails to capture the clear oscillations in the value of $g_{S,A}^{(2)}(t)$.

## Discussion

We note that oscillations resulting from the excitation of vibrational modes in different isotopes of $CCl_4$[54] were observed using ultrafast

stimulated Raman scattering in Ref. [42,44]. However, such experiments are well accounted for by a semi-classical model, in which the stimulated Stokes and anti-Stokes fields are classical. In the case of stimulated Raman scattering, the beating between the two pump laser fields is tuned resonant with the molecular vibration of interest, thereby driving a coherent collective vibration and resulting in a coherently oscillating Raman polarization in the sample, all of which behave as classical variables. In the Supplementary Methods, we show that our write and read pulses are not broad enough in frequency to stimulate Raman scattering from the vibrational modes at 655 cm$^{-1}$, and that classical coherence induced by possible low-frequency resonances can be excluded by measuring the delay dependence of the single-photon detection probabilities, as well as by simple considerations on the expected value of $g_{S,A}^{(2)}$.

In contrast, spontaneous Raman scattering and single photon counting demand a quantum description, in which the post-selected vibrational state naturally appears as a quantum coherent superposition involving all molecules coupled to the light field. Such a quantum model including experimental noise sources is detailed in the Supplementary Notes and solved numerically; here we present a simplified analytical model that captures the essence of the phenomenon and reproduces the observed quantum beats (cf. red solid line in Fig. 3).

We estimate that about $N \sim 10^{10}$ randomly moving molecules occupy the focal volume. If we consider them individually, the Raman interaction can be modeled as a tensor product of $N$ two-mode squeezing unitary operators $\hat{U}_k$ acting on the vacuum state of the Stokes photon ($S$) and the vibration ($v$)[20,52]. Assuming the same Raman cross-section for all the molecules and uniform field intensity within the focal volume, and ignoring corrections from time-ordering of operators[55,56] as valid in the low-gain regime, the state of the system after a short interaction is

$$|\psi\rangle = \hat{U}|\text{vac}\rangle$$
$$= \bigotimes_{k=1}^{N} \hat{U}_k |\text{vac}\rangle = \bigotimes_{k=1}^{N} e^{p\hat{A}_{S,k}\hat{A}_{v,k} - h.c.} |\text{vac}\rangle \qquad (1)$$
$$= |\text{vac}\rangle + \sqrt{p} \sum_{k=1}^{N} |1\rangle_{S,k} |1\rangle_{v,k} + O(p)$$

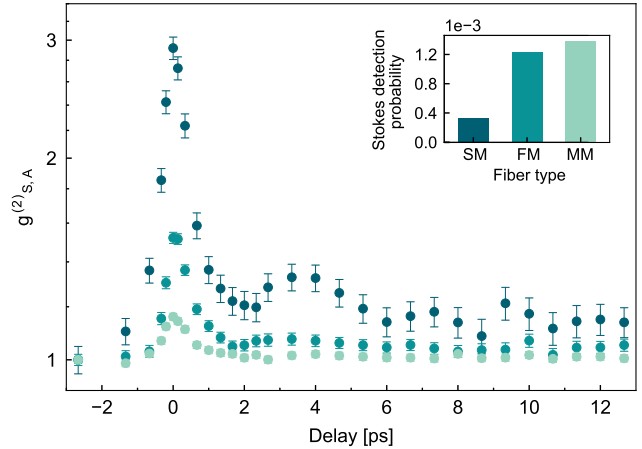

**Fig. 4 | Measured $g_{S,A}^{(2)}(t)$ using different optical fibers in collection path (see Fig. 2a).** From darker to lighter color: 3.5-$\mu$m core single-mode fiber (SM), 9-$\mu$m core telecom fiber (FM, few-mode), and 50-$\mu$m core multimode (MM) fiber. In the inset, we show the Stokes detection probability for the different fibers.

where $p \simeq 10^{-2}$ in our experimental conditions (low gain, spontaneous regime). $\hat{A}_{S,k}$ and $\hat{A}_{v,k}$ are the annihilation operators of Stokes photon and of the vibration, respectively, and $|1\rangle_{S,k}|1\rangle_{v,k} = \hat{A}_{S,k}^\dagger\hat{A}_{v,k}^\dagger|\text{vac}\rangle$.

After the detection of a Stokes photon, the post-selected vibrational state is given by

$$\rho_{ps}^{(j)} = \frac{\text{Tr}_v\left(\hat{K}_j\rho\hat{K}_j^\dagger\right)}{\text{Tr}\left(\rho\hat{K}_j^\dagger\hat{K}_j\right)} \qquad (2)$$

where $\rho = |\psi\rangle\langle\psi|$ and $\hat{K}_j$ are Kraus operators relative to a particular outcome $j$. A proper choice of Kraus operator is therefore key to model the experimentally observed vibrational state. Since the inverse duration of the laser pulses is much broader than the differences in vibration frequency between distinct molecules and since we collect the forward-scattered Raman signal through a single-mode fiber, information about which molecule is excited is erased, both spectrally and spatially. In this limit, the detection process can be described by a single Kraus operator $\hat{K} = |\text{vac}\rangle\langle\phi|$, where $|\phi\rangle = \frac{1}{\sqrt{N}}\sum_{k=1}^{N}|1\rangle_{S,k}$. Here, we have idealized the collective bright state as having zero momentum (only real coefficients).

Since the denominator of Eq. (2) is equal to $p$, it results that $\rho_{ps}(t=0) \equiv \rho_{ps}^{(1)} = |\psi_{ps}(0)\rangle\langle\psi_{ps}(0)|$ with

$$|\psi_{ps}(0)\rangle = \frac{1}{\sqrt{N}}\sum_{k=1}^{N}|1\rangle_{v,k}. \qquad (3)$$

From the cw Raman spectrum, we found four collective vibrational modes, which correspond to four sub-ensembles of molecules $i = 1, 2, 3, 4$ with frequencies $\Omega_i$ (in rad/s). Therefore, we can rewrite the post-selected state as

$$|\psi_{ps}(0)\rangle = \sum_{i=1}^{4}\beta_i|\chi_i\rangle_v \qquad (4)$$

where

$$|\chi_i\rangle_v = \frac{1}{\sqrt{N_i}}\sum_{k=1}^{N_i}|1\rangle_{v,k} \qquad (5)$$

is the state of a single collective excitation of the subensemble $i$, and $\beta_i^2 = N_i/N$ is the fraction of molecules in subensemble $i$. This quantity is equal to the fractional intensity of the corresponding cw Raman

peak as extracted from Ref. 48, i.e. $\beta_i^2 = 0.70, 0.03, 0.21$ and $0.06$. Therefore, the post-selected vibrational state at time $t$ after Stokes scattering is given by

$$|\psi_{ps}(t)\rangle = \sum_{i=1}^{4}\beta_i e^{-t/T_{2,i} - i\Omega_i t}|\chi_i\rangle_v \qquad (6)$$

where the coherence times $T_{2,i}$ are extracted from the spectrum in Fig. 1.

The probability of generating an anti-Stokes photon at time $t$ from this post-selected vibrational state is proportional to

$$P_A(t) \propto |\langle\psi_{ps}(0)|\psi_{ps}(t)\rangle|^2 = |\sum_{i=1}^{4}\xi_i(t)|^2 \qquad (7)$$

where $\xi_i(t) = \beta_i^2 e^{-t/T_{2,i} - i\Omega_i t}$. Here, we used the fact that the anti-Stokes detection geometry selects the same spatial mode as excited in Stokes scattering. We see that the Stokes–anti-Stokes correlations feature interference between the complex amplitudes $\xi_i(t)$, such that

$$g_{model}^{(2)}(t) - 1 = A_{tot}|\sum_{i=1}^{4}\xi_i(t)|^2 \qquad (8)$$

The complete fit function has the following expression:

$$F_{model} = g_{model}^{(2)} * f_{rise} + G_{FWM} \qquad (9)$$

where we convolute with a rise function and add a Gaussian function centered at zero delay that accounts for electronic FWM (which has vanishing memory compared to vibrational contributions). With this model we are able to fit the oscillatory decay of the heralded anti-Stokes intensity using only one scaling amplitude $A_{tot}$ as free parameter (plus a rise time $T_{rise}$ and the amplitude $A_g$ of the electronic FWM to fit the data close to zero delay), see Fig. 3. The value of $T_{rise}$ is longer than the nominal pulse duration due to dispersion from all the optical elements before the sample, which is not compensated for. The scaling factor $A_{tot}$ emerges naturally from the full quantum model solved numerically in the Supplementary Notes, once accounting for main sources of noise and inefficiencies.

While the quality of the fit provides strong support for our model, similar quantum beats would also result from coherence between energy levels of a single molecule[45–47]. In such a case, the exact detection geometry should have no impact on the observation, and in particular it would not be essential to post-select a single spatial mode to observe quantum beats. To test this hypothesis we repeat the same measurement using increasing fiber core sizes in the collection path, and therefore increasing number of spatial modes contributing to the total signal, with results shown in Fig. 4. The loss of visibility when measuring multiple optical modes provides further evidence that the observations are due to the spatial coherence generated through the experimental geometry and measurement post-selection. While a complete model for the multi-mode case is left for future work, we propose to interpret this result in light of information erasure: as the fiber core size increases, more and more information becomes available about the spatial origin of the signal, affecting the coherence of the post-selected state.

To summarize, we presented a time-domain measurement of spectrally resolved Stokes–anti-Stokes two-photon correlations on liquid $CS_2$ at room temperature, in the regime of spontaneous Raman scattering (Stokes scattering probability < 1%). Upon post-selection of events where both Stokes and anti-Stokes photons are collected through the same single-mode optical fiber, we observe quantum

beats that are perfectly consistent with a macroscopic quantum superposition of molecules sharing a single quantum of vibration. In particular, our experimental results are incompatible with the picture according to which spontaneous Raman scattering from a large ensemble is always the incoherent sum (statistical mixture) of single-molecule scattering events. Our experiment nourishes the debate about the relation between optical coherence and quantum coherence[57,58] and entanglement[59]. It questions whether optical coherent states are necessary to explain various forms of coherent Raman spectroscopy, for we do not stimulate the Raman process and yet observe coherent oscillations, as if we did. In the future, our photon counting approach can be adapted to probe inter- or intramolecular vibrational entanglement in more complex systems, as well as excitonic and vibrational polariton dynamics[60]. We also envision extensions of our work to probe how Raman scattering is affected by collective excitonic[61–64] or vibrational[65–70] strong coupling to a cavity, with implications for polariton chemistry[71].

## Methods

### Continuous-wave Raman spectroscopy

Liquid $CS_2$ (anhydrous, ≥99%, Sigma Aldrich) in a glass capillary tube is excited by a continuous-wave Ti:sapphire laser (Mira HP-F, Coherent) tuned at 780 nm with about 33 mW of power on sample. The Raman signal is detected with a high-resolution Shamrock SR-750 spectrometer equipped with a Newton 940 camera (Andor) and a 1800 l/mm, 400 nm blaze grating.

### Stokes–anti-Stokes correlation measurements

Liquid $CS_2$ is inserted in a quartz cuvette sealed with parafilm with ~ 0.2 mm wall thickness and ~ 1 mm optical path. The experiment is performed with the setup described in Fig. 2 (detailed scheme in Fig. S3 of the Supplementary Methods). The read and write pulses are generated by a Ti:sapphire oscillator (Tsunami, Spectra Physics, 80 MHz repetition rate) and a frequency-doubled optical parametric oscillator (OPO-X fs, APE Berlin), respectively. The Ti:sapphire wavelength can be tuned between 740 and 860 nm, while the OPO wavelength can be independently tuned between 505 and 740 nm. For the presented measurements, the Ti:sapphire wavelength is set at 810 nm and the OPO at 670 nm. The power is adjusted to generate no more than $10^{-2}$ Stokes photon per pulse within the spatial mode selected by the collection fiber, well below the onset of any stimulated Raman process, as confirmed by the linear power-dependence of both Stokes and anti-Stokes scattered intensities (see Supplementary Methods).

The Raman signal is sent either to a spectrometer equipped with a cooled CCD array or to a fiber-coupled single-photon counting module (SPCM-AQ4C, Excelitas) based on silicon avalanche photodiodes. The signal from the counting module is processed by FPGA-based correlation electronics (provided by TEDIEL S.r.l), which is interfaced by software to generate a coincidence histogram with a binning time of 1.67 ns. Such a histogram presents peaks each 12.5 ns, corresponding to the repetition period of the laser. When write and read pulses temporally overlap, a higher peak appears at zero delay of the coincidence histogram, corresponding to events where one photon is detected in each channel within the same write-read pulse sequence. The side peaks are due to uncorrelated photons. The number of coincidences for each peak is calculated by summing the counts of three adjacent bins. The number of coincidences in the zero-delay peak, divided by the average number of coincidences in the side peaks, is a measure of the normalized second-order two-photon correlation function. For these measurements, an average of the first 25 uncorrelated side peaks after the zero-delay peak is calculated, together with its standard deviation. Then, the error on the correlation function is propagated by assuming random fluctuations of the zero-delay peak (Poissonian statistics)[51].

## Data availability

The data that support the findings of this study have been deposited in a Zenodo repository with https://doi.org/10.5281/zenodo.7134740.

## Code availability

The code used in this study for data analysis and modeling has been deposited in a Zenodo repository with https://doi.org/10.5281/zenodo.7134740.

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

## Acknowledgements

The authors thank Vivishek Sudhir and Nicolas Sangouard for insightful discussions and valuable comments, Wen Chen for help with the cw Raman spectrum, and TEDIEL S.r.l for providing custom FPGA-based correlation electronics. This work has received funding from the Swiss National Science Foundation (SNSF) (projects no. 170684 and 198898) and the European Research Council's (ERC) Horizon 2020 research and innovation program (grant agreement no. 820196). A.P. acknowledges funding from the European Union's Horizon 2020 research and innovation program under the Marie Skłodowska-Curie grant agreement No. 754462.

## Author contributions

C.G. designed the experiment; V.V., S.T.V., and A.P. performed the measurements; V.V. and S.T.V. developed the theoretical model; all authors discussed the results and wrote the manuscript.

## Competing interests

The authors declare no competing interests.
