## [Peer Review File · Nature Communications]

REVIEWER COMMENTS

Reviewer #1 (Remarks to the Author):

The paper by Galland and coworkers reports on a very elegant experiment to probe the collective quantum coherence induced by spontaneous Raman (SR) scattering in a liquid. SR, in which a single "pump" field is scattered from a vibrational quantum, is typically considered an incoherent process, at odds with coherent Raman scattering (CRS) approaches, such as CARS and SRS, which involve excitation of a vibrational coherence by a pair of synchronized pump/Stokes light fields. This study investigates the quantum state of an ensemble of CS₂ molecules following SR scattering by means of a time-resolved two-photon Stokes-antiStokes coincidence measurement. Experimentally, a first ultrashort "write" pulse is inelastically scattered by a CS₂ solution and a second time-delayed "read" pulse interacts with the vibration giving rise to a spectrally distinct anti-Stokes signal. The Stokes-antiStokes correlation function $g^{(2)}_{S,A}(t)$ is then measured as a function of delay between write and read pulses, after the same spatial mode has been forced by a single-mode detection fiber. The results are quite spectacular, showing a clear oscillation indicative of a vibrational quantum beat or a collective vibrational coherence of the molecules interacting with light. This result is foundational for the understanding of the SR scattering process and as such of interest for a broad readership. It deserves, in my opinion, publication in Nature Communications.

I would only ask to the authors to provide the following minor clarifications:

- 1) The authors measure a maximum value of $g^{(2)}_{S,A}(0)$ of 3 as compared to an upper bound of 26, according to their estimates. Can they comment on the reasons for this discrepancy?
- 2) In their fitting model the authors use a rise time of $T_{rise} = 0.66$ ps, what is its physical origin?
- 3) The authors obtain their results on the rather simple triatomic molecule CS₂ with an elementary Raman spectrum. Can they comment on how they can be generalized to more complex molecules or to solids?

Reviewer #2 (Remarks to the Author):

The paper entitled "Measurement-Induced Collective Vibrational Quantum Coherence under Spontaneous Raman Scattering in a Liquid" by Valeria Vento, Santiago Tarrago Velez, Anna Pogrebna, and Christophe Galland reports the results on study of the Raman scattering in liquids. The focus was in quantum aspect of scattering and, in particular, the correlations between Stock and anti-Stock scattering.

The presented results are seemed to be of great interest, and the paper is deserved to be published.

The authors have demonstrated that spontaneous vibrational Raman scattering in the liquid phase and at room temperature can also induce a collective vibrational state in liquid carbon disulfide. The paper reports the measured time-resolved two-photon Stokes–anti-Stokes correlations mediated by the creation and annihilation of a vibrational quantum in the symmetric stretch mode of the molecules, and observe revivals after several picoseconds.

These quantum beats are signatures of a coherent superposition where a single vibrational quantum is shared between all molecules involved in Raman scattering and belonging to a spatial mode selected by the measurement geometry.

The reported results are consistent with the numerous works on emissive quantum memories using atomic ensembles that is important for the collective quantum coherence which can be induced by post-selection upon single photon detection – a key concept for quantum repeaters.

In conclusion, I am in full support to recommend the paper to be published because it reports the results that have important applications for quantum information processing and quantum computations.

Reviewer #3 (Remarks to the Author):

Vento et al. report signatures of vibrational coherence in liquid carbon disulfide when Stokes and anti-Stokes photons are sequentially emitted in the same spatial mode.

They believe the correlations they observe arise from spontaneous coherence associated with the emission of the Stokes photon. If correct, this would be a truly interesting result.

However, I strongly suspect that the effect they observe is induced by the ultrafast pump they use to "write." They do not provide sufficient information to eliminate this possibility, but the information

they do provide strongly suggests to me that they do impulsively generate a coherence with this pulse.

They report that their write pulse has a ~ 200 fs duration. Assuming this figure comes from an autocorrelation measurement, the true width is ~ 150 fs, with a 15 nm (230 wavenumber) bandwidth. The coherence generated by such a pulse would have 320 wavenumber FWHM, so resonances at 650 would be readily excited. Although the authors state that their "pulse intensities are far below the onset of any stimulated Raman processes," generating 0.01 Stokes photons per pulse in a single mode probably requires tens of pJ / pulse at 800 nm, which is enough to create a coherence in carbon disulfide (which has an exceptionally large third-order nonlinear coefficient).

Before the reported results can be seriously considered, the authors should 1) describe their laser power and focus conditions more completely and 2) determine and report the power dependence of their Stokes and anti-Stokes signals. If the coherence is created through spontaneous scattering, the scattered photon flux will be linear in the write pulse intensity. If the coherence is created impulsively by the write pulse, the Stokes and anti-Stokes photon flux will scale quadratically with the write pulse intensity.

Detailed replies to Reviewers' comments

Reviewer #1

- 1) *The authors measure a maximum value of $g^{(2)}_{S,A}(0)$ of 3 as compared to an upper bound of 26, according to their estimates. Can they comment on the reasons for this discrepancy?*

In theory we expect that $g^{(2)}_{S,A} \rightarrow 1/n_{th}$ for a bi-photon number state when limited by thermal noise in the phonon occupancy, which is quantified by the thermal phonon occupancy n_{th} [1]. However, other sources of noise further limit the maximum $g^{(2)}_{S,A}$: in particular spontaneous four wave mixing (of electronic origin) generated by the read pulse, which contributes to accidental counts. Finally, the detection efficiency and the dark counts are also sources of non-ideality. All these contributions are taken into account in our full quantum model (Supplementary Material) by choosing noise parameters that match the single-channel count rates measured on each detector while only one of the two pulses is active. This empirical approach has the benefit of accounting for all photon noise in excess of thermal noise (also leakage through filters, spurious scattering through the laboratory, etc.).

Change in main text, p. 8:

“Such a quantum model including experimental noise sources is detailed in the Supplementary Material and solved numerically”

- [1] M. D. Anderson, S. Tarrago Velez, K. Seibold, H. Flayac, V. Savona, N. Sangouard, and C. Galland, Two-Color Pump-Probe Measurement of Photonic Quantum Correlations Mediated by a Single Phonon, *Physical Review Letters* 120, 233601 (2018).

- 2) *In their fitting model the authors use a rise time of $T_{rise} = 0.66$ ps, what is its physical origin?*

$T_{rise} = 0.66$ ps was chosen to match the experimental data, and it physically accounts for all additional dispersion through the optics before reaching the molecules (all filters, a few lenses, the high-NA objective, the cuvette, etc.), as the pulse bandwidth is > 10 nm. The time resolution could certainly be improved with some experimental overhead by pre-compensating for dispersion, yet the time scales of interest in our experiment did not require such complications.

Change in main text, p. 10, we added the sentence:

“The value of T_{rise} is longer than the nominal pulse duration due to dispersion from all the optical elements before the sample, which is not compensated for.”

- 3) *The authors obtain their results on the rather simple triatomic molecule CS₂ with an elementary Raman spectrum. Can they comment on how they can be generalized to more complex molecules or to solids?*

In principle the complexity of the molecule is not an issue. Quantum beats are expected to emerge whenever multiple Raman-active modes have nearby frequencies lying within the pulse optical bandwidth, and a filter matching this bandwidth is used in detection. Also, we see no reason why the physics should be different in solids. Limitations would occur when material dispersion becomes large enough to create distinguishing information between nearby Raman peaks (by imprinting frequency information into time difference), which may happen in very thick samples (> 1 cm).

Reviewer #3

We thank Reviewer #3 for their discussion about possible origins of the spatial coherence we observe, and for proposing new measurements clarifying this origin. First, we recall the power regimes in which we are operating: the pump and the probe beams excite the sample through a 0.8-NA objective, so that the spot diameter in the focus is $\sim 1 \mu\text{m}$. For each measurement, the pump time-averaged power on the sample is in the range [5-7] mW, while the probe time-averaged power is in the range [35-45] mW, both at 80 MHz repetition rate (1 mW corresponds to 12.5 pJ pulse energy).

Next, we performed the experiments suggested by Reviewer 3:

If the coherence is created through spontaneous scattering, the scattered photon flux will be linear in the write pulse intensity. If the coherence is created impulsively by the write pulse, the Stokes and anti-Stokes photon flux will scale quadratically with the write pulse intensity.

According to the Reviewer's suggestion, we measured for different powers the Stokes intensity generated by the write pulse, and also the anti-Stokes intensity generated by the read pulse (which is close in frequency to the write pulse, so no significant difference in Raman cross section is expected). The integrated Stokes and anti-Stokes counts are plotted in Figs. 1(a) and (b) respectively. Clearly, the power dependence is linear for both of them, excluding a significant contribution of stimulated Raman scattering.

We believe these new data bring further support to our model, and fully address the Reviewer's concerns. In addition, they allow us to put a stringent upper bound on the mean number of Stokes photons generated by pulse (in the spatial mode we collect). Indeed, for the anti-Stokes signal, the crossover from linear to quadratic power dependence occurs when the mean Stokes photon number (and therefore generated phonon number) per pulse is on the order of the thermal occupancy, i.e. 0.04. We find no sign of quadratic increase up to 100 mW pump power. Therefore, in the conditions of the correlation measurements (~ 5 mW write power), the mean excited phonon number is expected to be on the order of $2 \times 10^{-3} \ll n_{\text{th}}$.

FIG.1. **(a)** Stokes intensity as a function of the write power on the sample. **(b)** Anti-Stokes intensity as a function of the read power on the sample. The counts are integrated over the relevant range of the Raman spectrum corresponding to the correlation measurements. The red lines are linear regressions.

We added these data and related discussion in the Supplementary Material and added the sentence in the main text, p. 5:

“[...] well below the onset of any stimulated Raman process, as confirmed by the linear power-dependence of both Stokes and anti-Stokes scattered intensities (see Supplementary Material)”

REVIEWER COMMENTS

Reviewer #3 (Remarks to the Author):

I thank the authors for providing the requested pulse information and performing the requested experiments. Their results suggest that they are not directly exciting a classical coherence of the 660 wavenumber Stokes and anti-Stokes modes.

However, having the pulse and focus parameters, I am even less convinced that classic coherence does not somehow play a role in the observed effect.

The authors correctly state that at 80 MHz, one obtains 12.5 pJ / mW average power. For a 0.8 NA focus (a $\sim 1 \mu\text{m}$ spot size), and a 200 fs pulse, the peak fluence of the pump and probe (write and read) pulses are $3 \times 10^{12} \text{ W/m}^2$ and $2 \times 10^{13} \text{ W/m}^2$ for 5 and 35 mW, respectively. This is ample pulse strength to generate coherence in CS₂.

In 1983, Green and Farrow observed a robust nonlinear signal (through the Kerr effect) at a much lower fluence of 10^{12} W/m^2 . (Greene & Farrow, *Chemical Physics Letters* 98, 273–276 (1983)).

In 1979, Compaan et al. (Coherent anti-Stokes Raman scattering with counterpropagating laser beams. *Opt Lett* 4, 170 (1979)), detected a CARS signal at the 660 wavenumber line in CS₂ with 20 kW peak power and roughly 0.07 NA focusing in a counterpropagating beam geometry. The fluence in the 1979 paper is roughly 10X less than the current authors use. Further, according to Compaan et al, the counterpropagating geometry yields three orders of magnitude less signal than the forward (co-propagating) geometry used by the authors of the current paper.

Given these two published results (and scores more using CS₂ to explore nonlinear effects), I am confident that the authors are generating a coherence. I am thus left wondering why they do not see nonlinear dependence on read and write beam intensity. I can only conclude that the autocorrelation of their pulses does not extend out to 600 wavenumbers, so they are not directly exciting those modes coherently. Knowing the spectrum of the pulses would confirm whether this is the case.

The oscillations they observe are at .25 THz, which is well within the bandwidth of the 200 fs pulse, so could be a direct result of a classical material coherence. There is a relaxation process in liquids at roughly this frequency, with roughly the same coherence time (part of the so-called "fast β " relaxation). Although, I am at a loss to explain why it would couple to correlations in Stokes and anti-Stokes emission.

In conclusion - The authors should at least cite some of the coherent Raman / CS2 literature and acknowledge that they are well within the nonlinear regime with their laser pulses. It seems that they are not coherently exciting Stokes and anti-Stokes bands at 660 wavenumbers, based on the intensity vs power. If they were, such coherence would lead to $g(2)_{S,A}$ oscillations at 660 wavenumbers, which they don't see.

Assuming the new results are accurate, I am satisfied that they are not directly coherently exciting the 660 wavenumber modes. However, it is unclear that any low-frequency material coherences (that they are certainly generating) also do not impact their results. Given that this is a previously unobserved result, the onus is perhaps on the authors to exclude other possible explanations. One possible experiment would be relatively straightforward - to find a liquid of similar molecular complexity with vibrational modes near 650 wavenumber but a much lower nonlinear coefficient and see if the $g(2)_{S,A}$ fringes become less visible. Propylene carbonate comes to mind as an option. The third-order nonlinearity is roughly 10X smaller.

propylene carbonate and essentially all liquids will have the β -fast relaxation at a similar frequency, but the oscillations in the response function observed here appear to be related to the frequency difference between the ostensibly coupled modes, and that would change with a different liquid.

We are grateful for the careful examination of our manuscript and our previous reply by Reviewer 3, leading them to conclude that they are “*satisfied that [we] are not directly coherently exciting the 660 wavenumber modes*”. We are convinced of the same and for the sake of completeness we repeat in the Appendix below the key aspects of our experiments that differ from the references cited by the Reviewer. In short, the main reason for which we do not coherently excite the Raman active modes around 660 cm^{-1} is that we are very far detuned from the resonance condition where two frequencies of the pump beams are spaced by 660 cm^{-1} . If such a condition is fulfilled we indeed observe strong coherent Raman signal, even at lower excitation power, but our scheme works precisely in the opposite regime. Each single pulse has a bandwidth of 190 and 80 cm^{-1} , respectively (and any tail potentially overlapping with the Raman signal is filtered before impinging on the sample), while the difference between the two pulses is $\sim 2 \times 580 \text{ cm}^{-1}$. Therefore, we are far detuned from the resonance condition for stimulated Raman scattering. More information can now be found in **Supplementary Figure S3 and associated discussion**.

We also appreciate the Reviewer’s effort to look for an alternative phenomenon that may explain our data, even though they eventually admit that they are “at a loss to explain why it would couple to correlations in Stokes and anti-Stokes emission.” We performed a thorough literature review regarding the cited beta-, fast-beta and related relaxation mechanisms in liquid, and also arrive at the conclusion that no such mechanism can explain our data, for several reasons detailed below. While we acknowledge that the Reviewer’s skepticism is a healthy position when facing unfamiliar results, we must stress that **our model is by far the simplest explanation for the data, as it makes no additional assumptions beyond the well-accepted and well-documented continuous-wave Raman spectrum**. In other words, from the mere knowledge of the experimental setting and the nature of the measurement any physicist will predict precisely the data we obtain (up to a scaling factor that depends on the exact signal-to-noise ratio, as we explain in the manuscript). Admittedly, second-order correlation measurements are not familiar to those working in the field of ultrafast spectroscopy.

As rightly pointed out in this editorial: Nature Nanotechnology **17**, p. 561 (2022), leading journals “ask reviewers to flag up [...] whether a simpler model or theory could explain the experimental data in a given manuscript”. We therefore feel obliged to emphasize that Occam’s razor was in our hands when analyzing the data, and we are absolutely convinced that no simpler model can exist for our data, for the good reason that we used zero extra assumption beyond the well-understood vibrational spectrum. As such, **any additional low-frequency resonance suggested by the Reviewer would have caused deviations from our model and failure to fit the data** (in exact ways which remain unknown, as admitted by the Reviewer).

While Occam’s razor should suffice to single out our model as the most natural and likely explanation for the data – given the photon-coincidence measurement performed – we underline additional reasons for which the mechanisms suggested by the Reviewer are unlikely to be valid competing explanations. First, after a careful literature review on beta-relaxation and related mechanisms, we were unable to find evidence for a low-frequency resonance around 0.25 THz in CS_2 at room temperature that would also exhibit a sufficiently long coherence time ($\sim 10 \text{ ps}$) to sustain the oscillations observed in our data. Rotational relaxation was measured to occur on a time scale of less than 2 ps [Greene, B. I.; Farrow, R. C. *Chem. Phys. Lett.* 1983, **98** (3), 273–276]. Broad resonance at very low frequency may occur close to a glass transition [Bertoldo Menezes, D *Polymer* 2016, **106**, 85–90; Zhao, Z. *J. of Chem.*

Phys. 2010, **132** (15), 154505] but here again we do not see any clear relation to our work.

Second, and most importantly, our measurement technique is designed so as to be insensitive to potential low frequency coherent oscillations. We measure the following quantity:

$$g^{(2)} = \frac{P(S \cap A)}{P(S)P(A)}$$

where $P(S \cap A)$ is the probability of coincident detection of a Stokes and anti-Stokes photons in the same pulse sequence, while $P(S)$ resp. $P(A)$ are the probability of single photon detection in each channel. Such photon correlation function is designed to reveal second-order coherence as predicted by a quantum mechanical description of light (and its interaction with matter). Let's assume that the "pump" beam generating the Stokes photon also induces any form of classical coherence affecting the subsequent probability of detecting an anti-Stokes photon. In such a scenario, the numerator factorizes as $P(S \cap A) = P(S) \cdot P(A)$ because the detection or non-detection of the Stokes photon is irrelevant to the induced coherence probed by the anti-Stokes signal (which is the case for a classical coherence). From the normalization of $g^{(2)}$ we find that it remains equal to one in this case, excluding a broad class of coherent effects. To make this point more obvious, **we now show in the Supplementary Figure S4 the anti-Stokes count rate vs. time delay**, in which the type of coherences discussed by the Reviewer would show up. However, this signal is dominated by laser power fluctuations and bares no signature of a beta-relaxation or related mechanism. On the contrary, the $g^{(2)}$ measurements shown in the main text are high reproducible and largely immune to laser power drift.

Photon correlation measurement is in many ways different from usual techniques in ultrafast spectroscopy. For example, in [Greene, B. I.; Farrow, R. C. *Chem. Phys. Lett.* 1983, **98** (3), 273–276], the Kerr effect is measured by monitoring a tiny birefringence induced by molecular orientation. Such effects typically lead to sub-percent level changes in polarization rotation. As can be seen from our data, the effects that we pick up are much larger, of order unity, as we measure a dimensionless quantity by design (we don't have to rescale our graph to "arbitrary units").

Finally, we thank the Reviewer for proposing new experiments on other molecular liquids. We totally agree that such experiments are exciting prospects for future work but they fall well beyond the scope of this article. For being valuable to the scientific community, this future study will be based on isotopically engineered molecules, featuring identical chemical properties but shifted Raman peaks. A well isolated Raman peak must be identified, which can then be shifted by controlled exchange of an isotope. Mixing the different isotopes in varying ratios will then enable to tune the visibility of the quantum beats, with the prospect of demonstrating an entanglement witness under maximal visibility, as proposed in the equivalent context of cavity optomechanics [Flayac and Savona *Phys. Rev. Lett.* 2015 **113**, 143603].

In summary, taking advantage of the valuable comments offered by the Reviewer, we performed the following modifications to the manuscript:

1. We added in the supplementary material a more detailed layout of the setup, in particular of the spectral filters and laser pulse bandwidths and central frequencies.

2. We added in the supplementary material the anti-Stokes count rates vs. time delay, showing no relevant signal contrary to expectations from a low-frequency resonance model.
3. We added a discussion in the supplementary material regarding possible low-frequency resonances in CS₂ that could be excited by the first pulse, and why we do not expect them to play any significant role in our measurements.

We hope that the revised manuscript will be deemed suitable for publication. As pointed out by Reviewers 1 and 2, our results will surely be received with great interest by the community; *Nature Communications* is the ideal journal for disseminating them and we are absolutely confident that the manuscript contains sufficiently strong and novel material at this stage to deserve publication, while motivating exciting future experiments in this new direction.

Appendix: additional responses to the Reviewer's comments

I thank the authors for providing the requested pulse information and performing the requested experiments. Their results suggest that they are not directly exciting a classical coherence of the 660 wavenumber Stokes and anti-Stokes modes.

However, having the pulse and focus parameters, I am even less convinced that classic coherence does not somehow play a role in the observed effect.

The authors correctly state that at 80 MHz, one obtains 12.5 pJ / mW average power. For a 0.8 NA focus ($a \sim 1 \mu\text{m}$ spot size), and a 200 fs pulse, the peak fluence of the pump and probe (write and read) pulses are $3 \times 10^{12} \text{ W/m}^2$ and $2 \times 10^{13} \text{ W/m}^2$ for 5 and 35 mW, respectively. This is ample pulse strength to generate coherence in CS₂.

*In 1983, Green and Farrow observed a robust nonlinear signal (through the Kerr effect) at a much lower fluence of 10^{12} W/m^2 . (Greene & Farrow, *Chemical Physics Letters* 98, 273–276 (1983).).*

Reply:

The experiment of Green and Farrow [1] measures a nonlinear response in CS₂ with time scales of 1.6 ps or shorter. These time scales are much shorter than any coherent process measured in our experiment (the lifetimes of the 4 modes are between 6.8ps and 18.4ps). In the 1983 experiment, two pulses of 200 fs at 625 nm (10 Hz repetition rate) from the same laser source are directed non-collinearly to a CS₂ sample. The pump induces a refractive index change, while the probe monitors the transient birefringence at variable time delays. The probe is attenuated by 30 times and it has 45° polarization with respect to the pump. After the sample, it passes through a polarizer at -45°. The probe is measured before and after the sample by two photodiodes connected to a lock-in amplifier in differential mode. Under a pump fluence of 10^8 W/m^2 , the result is a bi-exponentially decaying signal with a time constant of 1.56 ps attributed to rotational diffusion, plus a sub-picosecond time constant of no certain origin (but related to intermolecular light scattering by other authors). The exact value of the faster decay is not reported, but it can be estimated from the plot to be around 0.3 ps. By increasing the pump fluence to $1.2 \times 10^9 \text{ W/m}^2$ the temporal broadening of the measured signal obscures the fast decay, while the slow decay constant decreases from 1.56 ps to an even shorter value, such as 0.95 ps. Note also that this experimental technique measures the differences in refractive indices parallel and perpendicular to the pump pulse polarization, and therefore it probes relaxation mechanisms involving molecular re-orientation.

*In 1979, Compaan et al. (Coherent anti-Stokes Raman scattering with counterpropagating laser beams. *Opt Lett* 4, 170 (1979), detected a CARS signal at the 660 wavenumber line in CS₂ with 20 kW peak power and roughly 0.07 NA focusing in a counterpropagating beam geometry. The fluence in the 1979 paper is roughly 10X less than the current authors use. Further, according to Compaan et al, the counterpropagating geometry yields three orders of magnitude less signal than the forward (co-propagating) geometry used by the authors of the current paper.*

Reply:

In CARS, the anti-Stokes signal is coherently generated if the frequency difference between the two excitation pulses is tuned on the vibrational frequency of the

molecule. The Reviewer refers to the possibility of generating a CARS signal within a single pulse, i.e. with the two excitation beams being two spectral regions of the pump pulse separated by the vibrational frequency. In the work of Compaan and Chandra [2], a three-laser phase matching geometry is presented for CARS, where counterpropagating excitation beams allow for material-independent phase matching angle and good angular separation of the signal at the cost of reduced signal intensity. The excitation beams are chosen at 17452 cm⁻¹ (573 nm) and 16796 cm⁻¹ (592 nm) in order to target the CS₂ vibrational mode at 656 cm⁻¹. The two pump beams and probe beam have peak powers of 5, 5 and 10 kW, and they are focused in a length of 10 cm, 6.3 cm and 15 cm respectively

As shown in Fig. 2 of our manuscript, the FWHM of our pulse is below 80 cm⁻¹, much smaller than the Raman shift of any vibrational mode of CS₂. Furthermore, we use band pass filters in front of the detectors in order to select the Stokes and anti-Stokes emission from a spectral region of 20 nm around the modes at ± 655 cm⁻¹. The write pulse at 670 nm generates the Stokes signal (655 cm⁻¹) at ~ 701 nm, while the read pulse at 810 nm generates the anti-Stokes signal (-655 cm⁻¹) at ~ 769 nm. Therefore, in both cases, the detectors measure a spectral region that doesn't extend below 488 cm⁻¹. This means that, even if we suppose that a mode below 80 cm⁻¹ exists, the regions within a Raman shift of [-80, 80] cm⁻¹ from the write and read pulses, corresponding to [666, 674] nm and [805, 815] nm respectively, are widely excluded by our detection system. As a final note, even if the detection scheme and the sample specifics were setting us in a single-pulse CARS configuration, the comparison with the peak powers of Compaan et al. is not straightforward as suggested by the Reviewer. In fact, the peak power needed to enable the four-wave mixing process from a single pulse is larger than the one needed from two separate pulses at the right frequencies, because only fractions of the pulse far from the center are actually pumping the sample.

Given these two published results (and scores more using CS₂ to explore nonlinear effects), I am confident that the authors are generating a coherence. I am thus left wondering why they do not see nonlinear dependence on read and write beam intensity. I can only conclude that the autocorrelation of their pulses does not extend out to 600 wavenumbers, so they are not directly exciting those modes coherently. Knowing the spectrum of the pulses would confirm whether this is the case.

The oscillations they observe are at .25 THz, which is well within the bandwidth of the 200 fs pulse, so could be a direct result of a classical material coherence. There is a relaxation process in liquids at roughly this frequency, with roughly the same coherence time (part of the so-called "fast β " relaxation). Although, I am at a loss to explain why it would couple to correlations in Stokes and anti-Stokes emission.

As discussed in the previous point, we only measure the Stokes and anti-Stokes signals coming from a restricted spectral region around ± 655 cm⁻¹. If some other material coherence plays a role, it has to couple with the vibrational modes under investigation. A work from Menezes et al. [3] studies α , β and γ relaxation processes by Raman spectroscopy on nylon 6,6. The way these processes couple to vibrational motion is not completely clarified, but changes in the peak intensity, FWHM, and frequency, are found as a function of temperature. The three mostly affected vibrational modes are studied between -120°C to 120°C. In this temperature range, the C=O vibrational stretching mode undergoes a spectral shift no larger than 4 cm⁻¹, and FWHM fluctuations smaller than 2 cm⁻¹. The relaxation temperature for the β process is found at -30°C: at this temperature, the C=O stretching intensity increases of 50% with respect to room temperature. The β relaxation process happens close to

the glass transition. Given these results, we believe that the effect of this process on the selected CS₂ modes at room temperature is negligible, and in any case, it would just modify the relative contribution of the four modes to the quantum beats without questioning the validity of our model.

[1] Greene, B. I.; Farrow, R. C. *The Subpicosecond Kerr Effect in CS₂*. Chemical Physics Letters 1983, 98 (3), 273–276.

[2] Compaan, A.; Chandra, S. *Coherent Anti-Stokes Raman Scattering with Counterpropagating Laser Beams*. Opt. Lett. 1979, 4 (6), 170.

[3] Bertoldo Menezes, D.; Reyer, A.; Marletta, A.; Musso, M. *Determination of the Temperatures of the γ , β and α Relaxation Processes in Nylon 6,6 by Raman Spectroscopy*. Polymer 2016, 106, 85–90.

[4] Zhao, Z.; Huang, W.; Richert, R.; Angell, C. A. *Glass Transition and Fragility in the Simple Molecular Glassformer CS₂ from CS₂–S₂Cl₂ Solution Studies*. The Journal of Chemical Physics 2010, 132 (15), 154505.

[5] Moore, P.; Keyes, T. *Normal Mode Analysis of Liquid CS₂: Velocity Correlation Functions and Self-diffusion Constants*. The Journal of Chemical Physics 1994, 100 (9), 6709–6717.

REVIEWERS' COMMENTS

Reviewer #3 (Remarks to the Author):

The authors have formulated persuasive arguments in favor of their interpretation and have addressed my concerns. I congratulate them on a very nice manuscript.